# Adolescent Mental Health during the COVID-19 Pandemic: The Interplay of Age, Gender, and Mental Health Outcomes in Two Consecutive Cross-Sectional Surveys in Northern Italy

**DOI:** 10.3390/bs13080643

**Published:** 2023-08-01

**Authors:** Verena Barbieri, Giuliano Piccoliori, Angelika Mahlknecht, Barbara Plagg, Dietmar Ausserhofer, Adolf Engl, Christian J. Wiedermann

**Affiliations:** 1Institute of General Practice and Public Health, Claudiana College of Health Professions, 39100 Bolzano, Italyadolf.engl@am-mg.claudiana.bz.it (A.E.);; 2Faculty of Education, Free University of Bolzano, 39100 Bolzano, Italy; 3Department of Public Health, Medical Decision Making and Health Technology Assessment, University of Health Sciences, Medical Informatics and Technology—Tyrol, 6060 Hall, Austria

**Keywords:** adolescent health, mental health, COVID-19, age factors, gender differences

## Abstract

The coronavirus disease 2019 (COVID-19) pandemic has had a profound impact on the mental health and well-being of adolescents. This study aimed to investigate the development of health-related quality of life (HRQoL) and mental health among adolescents in Northern Italy by comparing cross-sectional surveys conducted in 2021 and 2022, with a particular focus on the influence of age and gender. The sample included adolescents aged 11–19 years from public schools in South Tyrol. Validated psychometric instruments were used to assess HRQoL, anxiety and depression symptoms, and psychosomatic complaints. Sociodemographic variables, COVID-19 burden, and pandemic-related factors were measured. Statistical analyses included chi-square tests, correlation coefficients, and logistic regression analyses. The results indicated that while the self-reported burden of adolescents decreased significantly in 2022 (*n* = 1885) compared to 2021 (*n* = 1760), there were no significant differences in symptoms of anxiety, depressive symptoms, low HRQoL, and increased psychosomatic complaints between the two surveys for both early and late adolescents. Females consistently had higher percentages for all outcome variables, and symptoms of anxiety increased with age in females, but not in males. Both genders experienced an increase in depressive symptoms and low HRQoL with age. The co-occurrence of mental health outcomes was observed, suggesting overlapping patterns among anxiety, depression, low HRQoL, and psychosomatic complaints. These findings underscore the intricate relationship between age, gender, and mental health outcomes among adolescents during the pandemic. It is important to recognize that late adolescents may exhibit distinct vulnerabilities and may require tailored support approaches to address their specific mental health challenges, differing from those needed for early adolescents.

## 1. Introduction

The coronavirus disease 2019 (COVID-19) pandemic has had a significant impact on the health-related quality of life (HRQoL) and mental health of individuals worldwide, including adolescents [1]. Before the (COVID-19) pandemic, adolescents in Europe generally had active social lives, with opportunities to socialize with peers in school, extracurricular activities, and social events; they were able to participate in sports, cultural events, and travel [2,3,4]. However, the pandemic has had a significant impact on adolescents’ lives [5,6,7]. Many schools were closed or transitioned to online learning, which disrupted their daily routines and limited their social interaction. The pandemic also led to the cancellation of many extracurricular activities, sports events, and cultural events, leaving adolescents with fewer opportunities to socialize and pursue their interests. Thus, the pandemic has caused psychological distress, including anxiety, depression, and loneliness, among adolescents [8,9,10]. They had to adapt to new ways of learning, socializing, and pursuing their interests while also dealing with significant psychological stressors. Many have also experienced a sense of loss or grief due to the changes brought about by the pandemic, such as the loss of social connections or missed milestones such as graduations or proms [11,12].

Social sciences and psychology differentiate the two age groups of adolescents. The term “early adolescence” is often used to refer to the period of development between ages 11–14 or 11–15, while the term “late adolescence” is used to refer to the period of development between ages 15–19 or 16–19 [13]. These terms are used to describe the different developmental stages that adolescents experience, including changes in physical, cognitive, emotional, and social development. Early adolescence is often characterized by the onset of puberty, increased self-consciousness, and a focus on peer relationships [14]. Late adolescence is characterized by increased independence, identity exploration, and future planning [15].

Differentiating between early and late adolescence may be important in studying the impact of the COVID-19 pandemic on HRQoL and mental health, for several reasons. Age-dependent differences can influence how adolescents experience and respond to the pandemic and its effects on psychosocial functioning [16,17,18]. For example, early adolescents may be more affected by disruptions to their education and social activities [19], while late adolescents may be more affected by disruptions to their career plans and future prospects [11,20]. These differences may require different interventions and supports to address the unique needs of each age group. Differentiating between early and late adolescence can help identify vulnerable subgroups within each age group, which may require targeted interventions and support. For example, research has shown that early adolescents from lower socioeconomic backgrounds may be at higher risk for mental health problems [21], whereas late adolescents who are in the process of transitioning to adulthood may experience increased stress and uncertainty [22].

Numerous longitudinal studies have been conducted to understand the impact of the pandemic on mental health and HRQoL among various population segments. For instance, nationwide longitudinal research in Germany highlighted significant decreases in HRQoL and increased mental health problems among children and adolescents during the pandemic [23,24,25]. Italy has also conducted longitudinal studies, although these have primarily focused on students aged 18 to 30 years [26] and youths with attention deficit hyperactivity disorder [27]. A notable gap remains in Italy where longitudinal studies specifically targeting adolescents are sparse. Most current studies focus on mental health, HRQoL, and psychosomatic complaints among adolescents during the pandemic, but they are predominantly cross-sectional in nature. These studies, while valuable, often suffer from inherent limitations that challenge the validity of conclusions drawn regarding the actual psychosocial impact of the pandemic on youth mental health in Italy.

The impact of the COVID-19 pandemic on HRQoL and mental health has been explored globally, including regions such as South Tyrol, Italy [28,29]. These investigations were largely driven by the need to understand the continuous development of children and adolescent mental health during the pandemic, particularly after the relaxation of restrictions and widespread vaccine distribution. Notably, it was found that despite the relaxation of restrictions from 2021 to 2022, symptoms of anxiety, depression, and psychosomatic complaints persisted. This persistence hints at either enduring effects of the pandemic or ongoing societal challenges. These studies also highlight discrepancies between parental and child reports of HRQoL and psychosomatic complaints, indicating the importance of obtaining multi-perspective insights [28,29].

Other studies demonstrate that isolation and loneliness increase the risk of mental health problems in young people during the COVID-19 pandemic and suggests that pandemic-related stressors and protective factors have unique associations with different aspects of adolescent adjustment during COVID-19 [30,31]. In line with these findings, our studies from South Tyrol also showed that certain psychosocial problems improved while others did not, highlighting that specific groups of children remain more at risk than others. Key risk factors were identified, such as low family climate due to the pandemic, single parenthood, higher parental education, and migration status [28,29].

The findings suggest that the percentages of heightened symptoms of anxiety, depression, and psychosomatic complaints will remain constant from 2021 to 2022. As a result, these symptoms can be viewed as either a long-term effect of the pandemic or a persistent societal challenge. The ‘lower family climate due to the pandemic’ emerged as the most significant predictor, followed by pandemic-related factors and social variables such as ‘parental workload due to the pandemic’, single parenthood, higher parental education, and migration status. However, the influence of ‘parental workload due to the pandemic’ had diminished in importance by 2022 [29].

The subsequent critical queries for participants classified as having elevated symptoms or those reporting low quality of life (i.e., ages 11–19) must now be addressed.
Age and gender have been identified as significant predictors for elevated cumulative scores for symptoms of generalized anxiety, depression, and psychosomatic complaints, as well as lower cumulative scores for health-related quality of life. These four scores can be categorized into normal and symptomatic cases. The aim was to explore whether the percentage of participants classified as symptomatic varied significantly by age and gender. This question was examined across four dichotomous outcome variables: symptoms of anxiety, depressive symptoms, at least three psychosomatic complaints per week, and low health-related quality of life.Is it feasible to infer one outcome variable from another? In other words, does any symptomatic variable serve as a predictor for other symptoms?Can adolescents with “lowered family climate due to the pandemic”, “difficult social factors not related to the pandemic”, or “increased burden due to the pandemic” be identified as vulnerable groups at a significantly higher risk for any of the four outcome parameters?

In this study, we aimed to explore the influence of the COVID-19 pandemic on HRQoL and mental health of early and late adolescents in South Tyrol. We conducted repeated cross-sectional surveys in 2021 and 2022. We will utilize validated psychometric tools to evaluate HRQoL and mental health outcomes by examining these results in relation to age and gender. This investigation is pivotal as it aims to shed light on the distinct challenges and experiences of young individuals in Northern Italy amidst the COVID-19 pandemic, thereby potentially informing the development of targeted interventions and support mechanisms to enhance the health and well-being of this demographic.

## 2. Materials and Methods

### 2.1. Research Framework and Participants

Our research on the impact of the COVID-19 pandemic on HRQoL and mental well-being among young individuals in South Tyrol is anchored in a design and participant pool outlined in an earlier publication [28]. For this iteration, we undertook a recurring cross-sectional study during the second and third years of the COVID-19 crisis. Strengthening the Reporting of Observational Studies in Epidemiology (STROBE), guidelines for reporting observational studies were followed [32]. The study adhered to the principles of the Declaration of Helsinki and received ethical approval from the Ethics Committee of the Autonomous Province of Bolzano, Italy (protocol code 0304767, protocol date 16 March 2022). Informed consent was secured from all participants, with adolescents signifying assent through voluntary online information provision and parents providing consent for their children’s participation.

In this study, we employ the term ‘Gender’ to denote the categories of ‘male’ and ‘female’. This is in line with the understanding that, while ‘Sex’ refers to biological differences, ‘Gender’ encapsulates the societal, psychological, and cultural differences between males and females. Our choice of terminology reflects the focus of our research on these psychological and social differences and their influence on HRQoL and mental health outcomes. It should be noted that while this study considers two genders—male and female—gender is not binary and can include other identities such as non-binary, among others. Our usage is guided by the principles outlined by the Canadian Department of Justice in their Guide to Gender-Based Analysis Plus (GBA+), recognizing the importance of distinguishing between ‘Sex’ and ‘Gender’ in research [33].

Data for our study, named ‘Corona and Psyche in South Tyrol’ (COP-S), were gathered through province-wide anonymous online surveys using SoSci Survey Software, version 3.2.46, in June 2021 and March 2022. This method mirrored the approach used in the COPSY German studies [24,25]. This study employed a population-based, cross-sectional design to assess adolescent mental health and health-related quality of life (HRQoL) in South Tyrol, Northern Italy. The sampling method entailed collaboration with the administration of public schools across South Tyrol. Invitations to participate were emailed to approximately 38,400 families with children attending public elementary, middle, or high schools. A reminder invitation was sent one week following the initial invite. Data collection was intentionally constrained to a short period to minimize potential impacts from changes in pandemic-related infection control measures. The parents of the students were invited to fill out the parent’s proxy version of the questionnaire for a child aged 7–19 years and proceed with the adolescent’s self-report for children aged 11–19 years. Details of the invitations, provided information, and questionnaires have been previously described [28].

This study focused on completed self-reports of adolescents aged 11 to 19 years, excluding proxy reports without matching self-reports in this age range. We also prioritized cases in which all four outcomes were completed, leading to a somewhat reduced sample size compared to the original study [29].

### 2.2. Assessment Tools

#### 2.2.1. Sociodemographic Information and COVID-19 Impact

We gathered sociodemographic details of the families and evaluated their experiences of the pandemic using the Comparative Analysis of Social Mobility in Industrial Nations (CASMIN) index [34,35] and the COPSY-Germany 2020 questionnaire [24].

#### 2.2.2. HRQoL and Mental Health

Details of the psychometric instruments are described elsewhere [28] and summarized here.

HRQoL was assessed using the KIDSCREEN-10 index, developed as a valid instrument for children and adolescents consisting of ten questions on physical, psychological, social, and school-dependent items presented on a 5-points response scale [36].

The self-reported Screen of Child Anxiety Disorders (SCARED) [37] with nine items on symptoms of generalized anxiety (GAD), represented on a 3-point response scale, was used as a psychometrically validated instrument for both German [38] and Italian [39] adolescents.

The presence and frequency of different psychosomatic problems of adolescents were assessed by parents and adolescents using the adopted Health Behavior in School-aged Children (HBSC) symptom checklist [40] within the last week with items presented on a 5-point response scale for quantification, and the results were compared with those of the COP-S 2021 survey.

The two item Patient Health Questionnaire-2 (PHQ-2) [41] on a 4-point Likert scale (0 to 3) was used as a depression screener, with a sum score of 3 as the cutoff. The PHQ-2 fulfills important psychometric criteria [42,43].

### 2.3. Data Analysis

Data interpretation followed the methodological path described in our previous work [28]. To comprehend the interrelationships among the four dichotomous outcome variables, we calculated sensitivities and specificities for all possible pairs. Logistic linear regression analyses were also executed for all classified symptomatic participants to control for sociodemographic and pandemic-specific risk factors and to account for the effects of the sum scores of other outcomes. Sample size calculations and statistical computations were performed using G-Power version 3.1.9.4 and SPSS version 25.0, respectively, adhering to recognized statistical thresholds of significance.

## 3. Results

### 3.1. Baseline Characteristics and Pandemic-Related Variables

Table 1 presents the baseline characteristics and pandemic-related variables for 2021 and 2022, respectively. Age, gender, migration background, and single parenthood were representative of the South Tyrolian population, according to the Provincial Institute for Statistics of South Tyrol (ASTAT) [https://astat.provinz.bz.it/de/datenbanken-gemeindedatenblatt.asp (accessed on 26 July 2023)].

Pandemic-related self-reported variables expressing burden due to the pandemic showed significantly lower percentages in 2022 than in 2021 (*p* < 0.001 for all variables). The outcome variables for the subgroups of the sociodemographic parameters are shown in Figure 1. For all four outcome variables, significantly higher percentages were detected for females in both survey years (*p* < 0.001 for each).

In both years, the outcome variables did not differ significantly between participants with and without a migration background. The percentage of depressive symptoms was significantly higher in the subgroup of participants with single parenthood than participants that were living with both parents in 2022 (*p* = 0.046), and in the subgroup of adolescents with at least three psychosomatic complaints a week (*p* < 0.001) in 2021.

The percentage of adolescents with at least three psychosomatic complaints a week was significantly lower in the subgroup of participants with low parental education than in the subgroup of participants with medium/high parental education in both 2021 (*p* = 0.021) and 2022 (*p* = 0.005). In the subgroup of participants with low parental education, the percentage of adolescents with low HRQoL was significantly smaller (*p* = 0.023) in 2021 than that in the subgroup of participants with medium/high parental education.

The subgroup of adolescents whose parents were dealing with mental health problems had a significantly higher incidence of low HRQoL by 2021 (*p* = 0.007). Similarly, a higher proportion of depressive symptoms was observed among adolescents with parents experiencing mental health problems (*p* = 0.029).

In both years, a significantly higher proportion of parents reported an increased workload due to the pandemic in groups exhibiting elevated levels for all four outcomes (*p* < 0.001 each).

### 3.2. Age- and Gender-Specific HRQoL and Mental Health Outcomes

The differences between the COP-S 2021 and COP-S 2022 surveys were not significant for symptoms of anxiety (26.8% vs. 27.5%), depressive symptoms (15.7% vs. 13.8%), low HRQoL (32% vs. 29.3%), or at least three symptoms of psychosomatic complaints (54.8% vs. 56.4%). Moreover, the comparison between 2021 and 2022, segmented by gender, revealed no significant changes in the percentage of elevated symptoms.

In both 2021 and 2022, females were significantly more likely than males to report symptoms of anxiety, depressive symptoms, low HRQoL and at least three psychosomatic symptoms per week. Specifically, in 2021, 34.2% of females reported symptoms of anxiety versus 19.0% of males (*p* < 0.001). Similarly, by 2022, the figures were 34.9% for females and 19.6% for males (*p* < 0.001). In terms of depressive symptoms, 20.5% of females reported such symptoms in 2021, compared to 10.7% of males (*p* < 0.001). By 2022, the figures were slightly lower, but the difference was still significant, with 17.5% of females and 9.9% of males reporting depressive symptoms (*p* < 0.001). Regarding low HRQoL, 37.1% of females reported this issue in 2021 versus 26.6% of males, and 33.1% of females versus 25.3% of males in 2022, with both differences being statistically significant (*p* < 0.001). Finally, for the reporting of at least three psychosomatic symptoms per week, the figures for females were 61.5% in 2021 and 62.3% in 2022, and 47.9% and 50.1% for males, respectively; these differences were also statistically significant (*p* < 0.001).

Figure 2 presents the percentages of reported symptoms of anxiety, depression, at least three psychosomatic complaints, and low HRQoL categorized by age, gender, and survey year. The data show a significant increase in symptoms of anxiety with age for females in both 2021 and 2022 (*p* = 0.001 and *p* < 0.001, respectively). However, this trend was not observed in males. Depressive symptoms showed a significant increase with age for both genders in both survey years (*p* = 0.001 and *p* = 0.002 for males and females in 2021, and *p* < 0.001 for both genders in 2022). The rate of self-reported low HRQoL also showed a significant increase with age for both genders in 2021 and 2022 (*p* < 0.001 for both genders in 2021, *p* = 0.003 for males and *p* = 0.001 for females in 2022). The frequency of reports indicating at least three psychosomatic complaints per week increased significantly with age among males in 2021 (*p* = 0.046) and among females in both survey years (*p* < 0.001).

### 3.3. Self-Versus Proxy-Reported HRQoL and Psychosomatic Complaints

Table 2 compares the percentages of low HRQoL and weekly occurrences of at least three distinct psychosomatic complaints, categorized by gender and differentiating between self-reports and proxy-reports for both 2021 and 2022. No significant changes were found between 2021 and 2022 in self-reported rates for either gender across the measured indicators. However, the proxy-reported rates displayed a significant reduction in the percentage of low HRQoL between 2021 and 2022 for both males (from 29% to 23.3%, *p* = 0.008) and females (from 31.6% to 23.9%, *p* < 0.001).

For males, the reports of low HRQoL did not significantly vary between self-reported and parent-reported data in either year. However, self-reported psychosomatic symptoms in 2022 were significantly higher than the proxy-reported symptoms (50.2% vs. 44.3%, *p* < 0.001).

In the case of females, self-reported low HRQoL was significantly higher than proxy-reported values, both in 2021 (37.1% versus 31.6%, *p* < 0.001) and 2022 (33.1% vs. 23.9%, *p* < 0.001). Additionally, the occurrence of at least three different psychosomatic symptoms was significantly more frequently reported in self-reports than in proxy-reports for females in both years (2021:61.5% versus 52.7%; 2022:62.3% versus 53.7%, respectively; *p* < 0.001).

### 3.4. Sensitivities, Specificities, and Correlations between HRQoL and Mental Health Outcomes

Often, mental health outcomes do not occur in isolation. For example, a person experiencing anxiety may experience symptoms of depression. Pairwise combinations allow us to understand the degree of overlap and co-occurrence between different outcomes. Appendix A presents pairwise combinations in terms of the percentages of the four self-reported main outcomes: anxiety, depression, at least three psychosomatic complaints per week, and low HRQoL.

Table 3 shows how well the presence of one outcome can predict the presence of another, providing insights into their interrelationships. The respective sensitivity and specificity of symptoms of anxiety, depressive symptoms, at least three psychosomatic complaints per week, or low HRQoL in the detection of another outcome in boys and girls are shown across the years 2021 and 2022. In addition, the table includes the Kendall’s tau-b correlation coefficient, which quantifies the degree of association between pairs of outcomes.

Starting with boys in 2021, the sensitivity of detecting symptoms of anxiety was 54.35% when depressive symptoms were present and vice versa was 30.67%, with specificities of 85.25% and 93.96%, respectively. The correlation between these outcomes was statistically significant (*p* < 0.001). This pattern is relatively steady in 2022, albeit with slightly decreased sensitivity and increased specificity for depressive symptoms.

For girls in 2021, the sensitivity of detecting symptoms of anxiety when depressive symptoms were present was higher than in boys (69.73%), and vice versa (41.75%), with specificities of 74.93% and 90.57%, respectively. In 2022, there was a notable increase in the sensitivity of detecting anxiety symptoms (79.53%), with minimal change in the other figures.

Regarding the relationship between symptoms of anxiety and HRQoL, for boys in 2021, the sensitivity of detecting symptoms of anxiety when low HRQoL was present was 43.23%, and vice versa was 60.74%. The specificities were relatively high, at 89.83% and 81.29%, respectively. These figures are relatively consistent in 2022. For girls, the sensitivities and specificities were slightly higher than those for boys, with some minor fluctuations from 2021 to 2022.

This table presents similar patterns of sensitivity, specificity, and correlation for outcomes related to depressive symptoms, psychosomatic symptoms, and HRQoL. In general, the sensitivity values tend to be lower than the corresponding specificity values, indicating that these outcomes are more specific but not as sensitively detected by the presence of other outcomes.

### 3.5. Construction of the Logistic Regression Model: Incorporating Pandemic-Related Factors and Health Outcomes

We created a separate logistic regression model for each survey year and gender. These models were designed to analyze how sociodemographic factors, pandemic-related conditions, and the outcomes themselves influenced the four binary outcomes.

#### 3.5.1. Predictors for the Model

The classification of all four outcomes into two categories (dichotomous categorization) showed significant variations based on age, gender, and increased parental workload due to the pandemic in both years. However, we found no significant variation based on migration background. Single parenthood, low parental education, and parental mental issues showed significant differences in some outcomes and years (Figure 1).

Minimal or no correlation was observed between sociodemographic factors and pandemic-related variables. With this in mind, age, gender and ‘either single parenthood, high parental education or parental mental health problems’ were included in the logistic regression model as binary predictors.

Building on prior research findings [28], we incorporated two additional binary predictors into the model: increased parental workload due to the pandemic and decreased family climate. These factors, along with other self-reported pandemic-related variables, such as increased school-related burden, reduced contact with friends, and increased usage of digital media, showed slight correlations with each other (Kendall-Tau-b between 0.1 and 0.3). They also highly correlated with the variable “children’s general burden due to the pandemic”. Given the high correlation, we excluded this variable from the regression model to avoid redundancy.

To streamline the model, we combined the remaining pandemic-related variables into a single composite score termed “children’s elevated burden”. This score encapsulates increased school-related stress, reduced social interaction, and enhanced digital media usage, serving as a comprehensive measure of pandemic-induced strain.

Finally, the sum scores for anxiety (SCARED/GAD), HRQoL (KIDSCREEN), depressive symptoms (PHQ-2), and number of self-reported weekly psychosomatic complaints ranging from 0 to 8 (HBSC) were included as independent predictors in separate models. These factors helped us explore the different aspects of children’s mental and physical health in relation to the pandemic.

#### 3.5.2. Logistic Regression Models for Dichotomous Outcomes

Logistic linear regression models highlighted the effects of the four outcome variables on each other in combination with the most significant predictors. Table 4 illustrates the model predictors of elevated anxiety symptoms, depressive symptoms, and low HRQoL among adolescents during the second and third years of the pandemic. The number of self-reported weekly psychosomatic complaints was an independent predictor. Key predictors across all categories included self-reported weekly psychosomatic complaints and lower family climate. For depressive symptoms and low HRQoL, older age was a significant predictor.

The association between predictors and outcomes exhibited variation based on gender and time. In the year 2021, boys demonstrated only a single significant predictor for anxiety symptoms, which was the number of psychosomatic complaints. Conversely, girls had two significant predictors: the number of psychosomatic complaints and the sum score of a pandemic-related burden. However, the importance of these predictors differed across outcomes.

In 2022, it was observed that parental workload significantly predicted boys’ symptoms of anxiety and low HRQoL. Moreover, girls’ symptoms of anxiety and depression were notably predicted by pandemic-related factors in both years. The model demonstrated a consistently higher fit for girls across all years and outcomes, suggesting a better prediction of girls’ symptoms or low HRQoL by the included factors.

Appendix A present additional results. For example, the anxiety sum score, when used as a predictor, showed significant associations with the same outcomes. Similarly, a higher depression symptom sum score also significantly predicted the outcomes.

As a general trend, low HRQoL was predicted by a diverse range of factors, with these models demonstrating the highest fit (Nagelkerke’s R^2^ > 0.5). For girls, symptoms of anxiety and depression could be predicted more effectively. Among these predictive factors were psychosomatic complaints, low family climate, and the respective other symptoms, with the latter being combined with age for depression, and not for anxiety.

Psychosomatic complaints were most effectively predicted by symptoms of anxiety, in combination with age or depressive symptoms. No model using HRQoL as a predictor has been presented until now. However, in the calculation of these models, HRQoL emerged as the only significant predictor, which made the calculation of sensitivity and specificity sufficient.

Taken together, these findings underscore the complex interplay between psychosomatic complaints, depressive and anxiety symptoms, family climate, age, and the burden of the pandemic in shaping adolescents’ well-being.

In Appendix A, gender differences are also evident, with girls generally showing higher odds ratios associated with low HRQoL, depressive symptoms, and psychosomatic complaints, especially in 2021. For boys, low HRQoL showed a marked increase over the two years, with family climate and anxiety being significant contributors. The data revealed the pervasiveness of the anxiety sum score as a predictor, reinforcing the strong relationship between anxiety and other health outcomes across both genders and years.

As shown in Appendix A, including self-reported symptoms of depression as an independent predictor, age was a significant factor for low HRQoL in both boys and girls across both survey years, with older adolescents showing higher odds ratios. In terms of gender differences, girls generally had higher odds ratios associated with low HRQoL and psychosomatic complaints than boys. For elevated symptoms of anxiety, boys showed a slight but significant inverse relationship with age by 2022. Additionally, the impact of depressive symptoms was pervasive across all outcomes and both genders, indicating a strong relationship between depressive symptoms and the other issues examined.

The tables highlight the complexity of mental health outcomes among adolescents during the pandemic, with age, gender, psychosomatic complaints, and mental health symptoms playing varying roles over time.

## 4. Discussion

The reported demographic features and pandemic-related factors, which included age, gender, migration background, and single parenthood, were representative of these factors in the South Tyrolian population. The pandemic-related self-reported burden of adolescents showed a significant reduction in 2022 compared to that in 2021. However, there were no significant differences between the COP-S 2021 and COP-S 2022 surveys for symptoms of anxiety, depressive symptoms, low HRQoL, and at least three symptoms of psychosomatic complaints per week. Among the surveyed groups, females had significantly higher percentages of all four outcome variables. There were no significant differences between the participants with and without migration backgrounds. However, adolescents with single parents and those with parents dealing with mental health problems had a higher incidence of certain mental health outcomes. Furthermore, there was a significant increase in symptoms of anxiety with age in female adolescents but not in male adolescents. Depressive symptoms and low HRQoL increased significantly with age in both the genders. There were no significant changes in self-reported rates from 2021 to 2022 for either gender, but the proxy-reported rates showed a significant reduction in the percentage of low HRQoL for both genders. Mental health outcomes often co-occur, indicating a significant overlap between outcomes, such as symptoms of anxiety, depressive symptoms, low HRQoL, and psychosomatic complaints. Boys and girls exhibited different patterns of overlap, and the sensitivity of detecting one outcome when another was present varied.

Overall, these results demonstrate complex interrelationships between different mental health outcomes, but clearly showed that they did not change from 2021 to 2022, as confirmed in other studies from Sweden [44], the USA [45], and Iceland [46]; they showed a slight decrease in Germany [47]. A Japanese study [48] showed a post-pandemic increase in mental disorders.

Several reviews and studies have examined the short and long-term effects of the COVID-19 pandemic on the mental health of young people [49,50,51,52,53,54,55,56,57,58,59,60,61,62,63,64]. Bussières et al. [52] found that mental health problems in children up to 13 years of age increased three times more in European countries than in Asian countries. Other studies [57,63,65] have also highlighted an increase in mental health problems among children and adolescents, particularly in Europe. A comparison between pre-pandemic and pandemic periods revealed an increase in depressive symptoms among European youth [57], and severe increases in mental disorders have been reported in Germany, Italy, and Poland [63].

Our study provides several key insights into the impact of various factors on adolescents’ mental health during the second and third year of the pandemic. First, late adolescence plays a significant role in influencing health outcomes, particularly depressive symptoms [66] and low HRQoL. The influence of age appears to be more pronounced in later stages of the pandemic. Second, differences were observed between boys and girls. Psychosomatic complaints were more influential on boys’ anxiety levels and depressive symptoms, while family climate and socio-demographic conditions contributed significantly to low HRQoL and depressive symptoms in girls. Third, psychosomatic complaints consistently predicted anxiety and low HRQoL across all outcomes, ages, and genders (Appendix A and Table 4), with Appendix A highlighting their significance in predicting low HRQoL and elevated anxiety symptoms. Lastly, anxiety scores significantly predicted low HRQoL in both boys and girls, whereas depressive symptoms significantly predicted low HRQoL.

The data suggest a strong relationship between the various mental health outcomes in both boys and girls. When one outcome is present, such as anxiety, it is often associated with the presence of another, such as depressive symptoms or poor HRQoL. However, the degree and direction of this association can vary according to gender and year. A stronger relationship between different pandemic mental health problems than between different pre-pandemic mental health problems has been reported [67]. These findings indicate that pandemic-related factors affect mental health outcomes. In our study, the most important health-related factor was a lower family climate due to the pandemic, followed by pandemic-related adolescent burdens.

While the distinction between early and late adolescence is not directly provided in the data tables, we can infer the potential impacts based on our understanding of adolescent development and the observed effects. Age has emerged as a significant predictor of mental health outcomes, particularly depressive symptoms and low HRQoL. Similar findings have been previously reported [68]. Late adolescence is characterized by substantial changes and challenges such as increased responsibility, educational, or occupational transitions, and the development of complex social and emotional relationships. These challenges, combined with the ongoing stress and uncertainty of the pandemic, could explain the heightened vulnerability to depressive symptoms and the reduced HRQoL observed in older adolescents.

Gender has emerged as the most important factor in predicting mental health problems in adolescents. Females were more affected across all outcome variables, and increasing age was a significant factor for all outcomes in the female subgroup [69,70,71,72]. Age, however, was not a significant predictor of anxiety symptoms in boys. The observed gender differences may be related to the different stages of adolescence. Girls tend to enter puberty earlier than boys, which could influence the earlier onset of certain mental health issues such as depression. Sociodemographic and family climate factors appear to have a more significant impact on girls’ mental health outcomes. The higher rate of psychosomatic complaints reported by boys may be associated with societal expectations of masculinity, which discourage emotional expression, leading to physical symptoms.

The significance of psychosomatic complaints as a predictor across all outcomes, ages, and genders suggests that their potential increases with age. As adolescents mature, they become more aware of their bodies and emotions, which could lead to increased reporting of psychosomatic symptoms. This can be exacerbated in the stressful context of the global pandemic.

Regarding anxiety symptoms, the regression models identified “pandemic-related burden of adolescents” as a significant predictor for girls but not for boys. In the prediction of depressive symptoms, this factor was significant only for girls in 2021. Another noteworthy pandemic-related factor was the “parents’ extended workload due to the pandemic”, which was found to be more important in 2021 than in 2022, specifically in the subgroup of boys. Lower family climate due to the pandemic was consistently found to be a significant predictor across all models, aligning with findings from other European studies where school closures and family difficulties were predictive factors for increased mental health problems during the pandemic [57,61].

In summary, these findings emphasize the importance of considering the different stages of adolescence when examining the impacts of significant life events, such as the pandemic, on mental health outcomes. Early and late adolescence are distinct developmental periods, and adolescents’ responses to stressors and their mental health impacts can vary. The pandemic likely amplified these differences, highlighting the need for age- and developmentally appropriate mental health support for adolescents.

Geweniger et al. [73] found a strong association between parent-reported child mental health problems and pandemic-related variables, such as increased family conflict, inadequate social support, and caregivers’ mental health. Similar observations were made in German-speaking countries (Austria, Germany, Liechtenstein, and Switzerland), where a substantial proportion of children and adolescents experienced pandemic-related mental health problems [74,75,76]. In Italy, depression in children was found to be dependent on parental stress [77].

The use of the terms “early adolescence” and “late adolescence” enables researchers and practitioners to specifically target interventions and tailor approaches to meet the unique needs of adolescents at each developmental stage. This supports the healthy development and well-being of young people, as they navigate the challenges and opportunities of adolescence. Although the distinction between early and late adolescence may not have directly influenced pandemic management as policies primarily focused on public health, it is crucial to consider the developmental needs and experiences of early and late adolescence when developing policies and interventions related to education, mental health, and social support. For instance, school closures and transitions to online learning have significantly affected the educational experiences of all adolescents. However, younger adolescents may require more support and supervision to successfully navigate online learning because of their developmental stage. Early adolescents may need more support in managing peer relationships and social development, whereas late adolescents may require assistance in managing transitions to adulthood and planning for the future. It is also important to take gender into account when using these terms. The data reveal that pandemic-related burdens were experienced as more self-related by girls and more parent-related by boys.

Logistic regression models revealed that psychosomatic complaints played a significant role in predicting depressive symptoms in girls, with a higher number of complaints associated with a greater likelihood of experiencing depression. When combined with increasing age and low family climate, the probability of developing depressive symptoms is further heightened. Anxiety symptoms in girls, along with higher age and a lower family climate, also contribute to the likelihood of experiencing depressive symptoms, and the significance of Nagelkerke’s R^2^ values is reduced for both genders. However, in the 2022 data, a combination of psychosomatic complaints, depressive symptoms, low family climate, and pandemic-related burden emerged as reasonably good predictors of anxiety symptoms in girls (Nagelkerke’s R^2^ > 0.35). Interestingly, age no longer remained significant in the model, indicating that distinguishing between early and late adolescence is less crucial for predicting anxiety symptoms.

Previous research in South Tyrol, Italy [28], indicated positive HRQoL and mental health outcomes among young people. It is important to identify subgroups that may be disproportionately affected by the pandemic. Although socioeconomic factors are deemed to be of secondary importance in predicting mental health and quality of life among South Tyrolean adolescents, single parenthood has emerged as a significant factor. Adolescents living with a single parent during the pandemic reported higher levels of depressive symptoms and lower HRQoL than those living with both parents. While the significance of “low family climate” outweighs other factors across all outcome variables, further investigation is needed. Understanding the impact of a positive family climate on adolescents’ mental health is also crucial. Similar effects of family climate on adolescent mental health have recently been reported [47,78].

In summary, the findings highlight that late adolescent girls who experience a higher number of psychosomatic complaints and encounter social difficulties with parents and friends are particularly prone to experiencing depressive symptoms, often accompanied by anxiety symptoms. Age appeared to be less significantly related to anxiety symptoms. These results underscore the importance of psychosomatic complaints as an indicator of mental health problems, particularly in older girls.

Our findings confirmed that HRQoL is influenced by multiple factors beyond mental health problems, making it variable and dependent on daily circumstances. Logistic regression models demonstrated that adolescents’ low HRQoL can be predicted by factors such as low family climate, children’s burden due to the pandemic, age, and symptoms (anxiety, depression, and psychosomatic). Extended parental workload during the pandemic was a significant factor for boys. Although health-related quality of life improved with the easing of restrictions, pandemic-related closures and parameters still had lower values. This indicates that HRQoL is a dynamic construct that is influenced by daily mood, social factors, and mental well-being. Additionally, the prevalence of low HQoL scores increased with age in girls.

This study has some limitations. (i) Our study focused solely on adolescents in South Tyrol, Italy, which limits the generalizability of the findings to other populations. The specific sociocultural context and healthcare system of South Tyrol may have influenced the mental health outcomes observed. (ii) The study relied on cross-sectional data collected at two time points (2021 and 2022), which limits our ability to establish causal relationships or assess long-term effects. (iii) while our study examined various factors, such as age, gender, migration background, and single parenthood. Other important variables that could influence mental health outcomes, such as socioeconomic status and access to mental health services, have not been fully explored. Considering a broader range of determinants may provide a more comprehensive understanding of the complex factors influencing adolescent mental health during times of crises. (iv) Our study relied on a specific set of measures and did not capture the full spectrum of mental health outcomes and experiences of the adolescents. Other dimensions, such as resilience, coping strategies, and social support, have not been explored extensively.

Despite these limitations, our study provides valuable insights into the mental health outcomes of adolescents in South Tyrol during the COVID-19 pandemic.

## 5. Conclusions

Despite a significant reduction in the self-reported burden of adolescents in 2022 compared to 2021, there were no significant differences in symptoms of anxiety, depressive symptoms, low HRQoL, and psychosomatic complaints between the two survey years. Females consistently exhibited higher percentages of all four outcome variables. While the migration background did not show significant differences, single parenthood and parents dealing with mental health problems were associated with a higher incidence of certain mental health outcomes. Symptoms of anxiety increased with age in females but not in males, whereas depressive symptoms and low HRQoL increased with age in both genders. The co-occurrence of mental health outcomes suggests a complex interrelationship between anxiety, depression, low HRQoL, and psychosomatic complaints. Boys and girls displayed different patterns of overlap and the sensitivity of detecting one outcome when another was present varied. Our findings emphasize the need for tailored mental health support for adolescents based on age, gender, and specific mental health challenges. It is important to consider the developmental stages of early and late adolescence when addressing the impact of significant life events such as the pandemic. Furthermore, the presence of psychosomatic complaints emerged as a significant predictor of depressive symptoms in girls, while positive family climate showed consistent significance across all outcome variables. These findings highlight the importance of ongoing research to understand the nature of mental health outcomes and influence of family dynamics. Future interventions should address adolescents’ unique needs, particularly in relation to their mental health and well-being.

## Figures and Tables

**Figure 1 behavsci-13-00643-f001:**
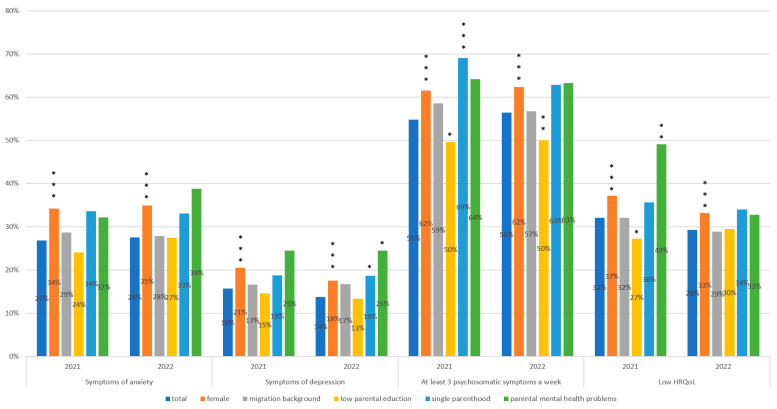
Four dichotomous outcome variables (symptoms of anxiety, depressive symptoms, at least three psychosomatic complaints a week, low HRQoL) in relation to sociodemographic characteristics of adolescents (*n* = 1760 in 2021; *n* = 1885 in 2022). * *p* < 0.05, ** *p* < 0.01, *** *p* < 0.001 (Chi square test).

**Figure 2 behavsci-13-00643-f002:**
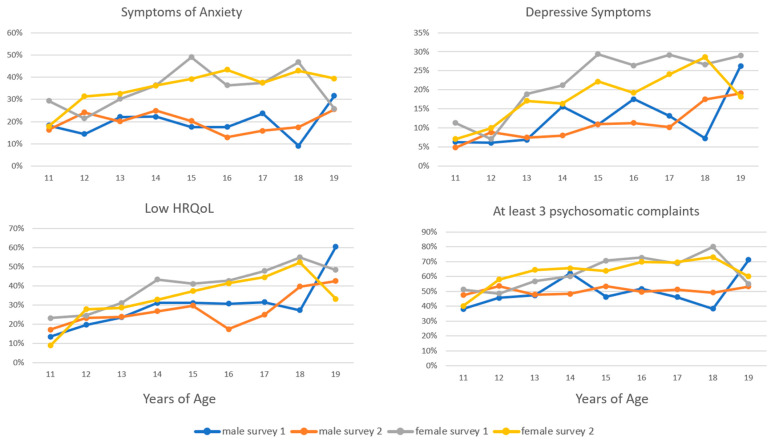
Percentages of self-reported elevated symptoms of anxiety (SCARED), depression (PHQ-2), at least three psychosomatic complaints (HBSC), and low HRQoL (KIDSCREEN) per year of age, gender, and survey (survey 1, COP-S 2021; survey 2, COP-S 2022).

**Table 1 behavsci-13-00643-t001:** Sociodemographic and pandemic-related characteristics of adolescents aged 11–19 years in COP-S 2021 and COP-S 2022 self-report samples.

	Total ^†^	
	2021 (*n* = 1760)	2022 (*n* = 1885)
Sociodemographic	M (SD)	M (SD)
Age	14.22 (2.34)	14.40 (2.31)
	*n* (%)	*n* (%)
Female	903 (51.3)	975 (51.7)
Migration background ^‡^	181 (11.1)	179 (10.3)
Low parental education ^‡^	419 (24.8)	391 (21.5)
Single parenthood ^‡^	149 (8.6)	188 (10.0)
Parental mental health problem ^‡^	53 (3.0)	49 (2.6)
Pandemic-related		
Extended parental workload ^‡,§^	605 (38.1)	619 (35.6)
General burden	528 (30.2)	22.9 (9.58)
Lower family climate	464 (26.5)	348 (18.5)
Elevated burden at school	1151 (65.5)	1087 (57.9)
Less contact with friends	1067 (60.7)	720 (38.2)
Extended use of digital media	1214 (69.1)	1049 (55.9)

**^†^** Participants who completely answered the four outcome questionnaires (anxiety, depression, at least three psychosomatic complaints per week, and HRQoL). ^‡^ Proxy-reported. ^§^ Worse compared to before the pandemic.

**Table 2 behavsci-13-00643-t002:** Comparison of low HRQoL and weekly occurrence of at least three psychosomatic complaints in adolescents during the second and third year of the COVID-19 pandemic. Stratification by gender, self-reports versus proxy reports, including all complete cases aged 11–19 years.

Gender	COP-S Survey	*n*	Low HRQoL ^†^			At least 3 Psychosomatic Complaints a Week ^†^		
			Self-Reported	Proxy-Reported	*p-*Value ^‡^	Self-Reported	Proxy-Reported	*p-*Value ^‡^
Boys	2021	858	229 (26.7)	244 (29.0)	n.s.	411 (47.9)	377 (44.8)	n.s.
	2022	911	231 (25.4)	208 (23.3)	n.s.	457 (50.2)	395 (44.3)	<0.001
	*p*-value ^§^		n.s.	0.008		n.s.	n.s.	
Girls	2021	903	335 (37.1)	280 (31.6)	<0.001	555 (61.5)	465 (52.7)	<0.001
	2022	975	323 (33.1)	228 (23.9)	<0.001	607 (62.3)	511 (53.7)	<0.001
	*p*-value ^§^		n.s.	<0.001		n.s.	n.s.	

**^†^** Low HRQoL according to KIDSCREEN and at least three psychosomatic complaints according to HBSC. **^‡^** McNemar-Test. ^§^ *p*-values resulting from the χ^2^ test comparing the results of 2021 and 2022 with low vs. normal/high HRQoL and at least three different psychosomatic complaints a week of self-reported and proxy-reported HRQoL and at least three different psychosomatic complaints per week, respectively. n.s., not significant.

**Table 3 behavsci-13-00643-t003:** Gender-based comparative analysis of HRQoL and mental health outcomes of sensitivity, specificity, and interrelationship in COP-S 2021 and COP-S 2022 data.

Gender	COP-S Year	Outcome to Detect	When Having Other Outcome		Kendall Tau-b Correlation Coefficient
			Sensitivity	Specificity	
Boys	2021	Anxiety	54.35 [43.63; 64.78]	85.25 [82.54; 87.69]	0.312 ***
		Depression	30.67 [23.70; 38.37]	93.96 [91.92; 95.61]	
	2022	Anxiety	52.22 [41.43; 62.87]	84.04 [81.36; 86.48]	0.273 ***
		Depression	26.40 [20.09; 33.52]	94.13 [92.18; 95.72]	
Girls	2021	Anxiety	69.73 [62.56; 76.25]	74.93 [71.59; 78.06]	0.380 ***
		Depression	41.75 [36.19; 47.47]	90.57 [87.93; 92.80]	
	2022	Anxiety	79.53 [72.70; 85.31]	74.63 [71.47; 77.60]	0.432 ***
		Depression	40,00 [34.75; 45.42]	94.49 [92.42; 96.13]	
Boys	2021	Anxiety	43.23 [36.72; 49.92]	89.83 [87.19; 92.08]	0.373 ***
		HRQoL	60.74 [52.79; 68.28]	81.29 [78.19; 84.13]	
	2022	Anxiety	46.75 [40.18; 53.41]	89.71 [87.17; 91.89]	0.400 ***
		HRQoL	60.67 [53.09; 67.90]	83.22 [80.31; 85.86]	
Girls	2021	Anxiety	58.51 [53.03; 63.84]	80.11 [76.58; 83.31]	0.393 ***
		HRQoL	63.43 [57.57; 68.09]	76.60 [72.98; 79.95]	
	2022	Anxiety	66.06 [60.64; 71.17]	80.66 [77.45; 83.61]	0.463 ***
		HRQoL	62.94 [57.57; 68.09]	82.83 [79.67; 85.69]	
Boys	2021	Anxiety	32.36 [27.86; 37.12]	93.29 [90.56; 95.43]	0.327 ***
		Psychosom	81.6 [74.78; 87.22]	60 [56.25; 63.66]	
	2022	Anxiety	30.84 [26.65; 35.31]	91.85 [88.94; 94.20]	0.286 ***
		Psychosom	79.21 [72.51; 84.92]	56.89 [53.21; 60.51]	
Girls	2021	Anxiety	49.19 [44.95; 53.43]	88.35 [84.24; 91.71]	0.398 ***
		Psychosom.	88.35 [84.24; 91.71]	52.53 [48.42; 56.60]	
	2022	Anxiety	50.16 [46.15; 54.18]	90.84 [87.43; 93.57]	0.419 **
		Psychosom	90.76 [87.33; 93.52]	63.60 [60.22; 66.89]	
Boys	2021	Depression	32.61 [26.59; 39.08]	97.16 [95.54; 98.31]	0.421 ***
		HRQoL	80.43 [70.85; 87.97]	79.77 [76.74; 82.56]	
	2022	Depression	30.74 [24.85; 37.12]	97.24 [95.72; 98.33]	0.407 ***
		HRQoL	78.89 [69.01; 86.79]	80.51 [77.63; 83.17]	
Girls	2021	Depression	46.59 [41.16, 52.07]	95.10 [93.00; 96.72]	0.502 ***
		HRQoL	84.86 [78.87; 89.70]	75.21 [71.88; 78.33]	
	2022	Depression	44.95 [39.48; 50.52]	95.92 [94.12; 97.30]	0.506 ***
		HRQoL	84.80 [78.52; 89.82]	77.86 [74.83, 80.69]	
Boys	2021	Depression	18.45 [14.82; 22.54]	96.23 [94.03; 97.79]	0.223 ***
		Psychosom	81.52 [72.07; 88.85]	56.14 [52.54; 59.69]	
	2022	Depression	17.38 [14.05; 21.33]	97.62 [95.78; 98.81]	0.264 ***
		Psychosom	90.00 [81.86; 95.32]	54.20 [50.72; 57.65]	
Girls	2021	Depression	31.13 [27.31; 35.15]	96.86 [94.45; 98.42]	0.340 ***
		Psychosom	94.05 [89.12; 96.99]	46.94 [43.23; 50.66]	
	2022	Depression	27.51 [24.02; 31.21]	98.92 [97.26; 99.71]	0.337 ***
		Psychosom.	97.66 [94.12; 99.36]	45.27 [41.79; 48.79]	
Boys	2021	HRQoL	45.26 [40.37; 50.21]	90.38 [87.26; 92.95]	0.402 ***
		Psychosom	81.22 [75.55; 86.06]	64.23 [60.34; 67.98]	
	2022	HRQoL	42.01 [37.44; 46.69]	91.41 [88.44; 93.82]	0.384 ***
		Psychosom	83.12 [77.65; 87.71]	61.03 [57.25; 64.71]	
Girls	2021	HRQoL	54.23 [49.99; 58.44]	90.23 [86.62; 93.14]	0.448 ***
		Psychosom	89.85 [86.11; 92.87]	55.28 [51.09; 59.42]	
	2022	HRQoL	49.26 [45.21; 53.31]	93.48 [90.45; 95.78]	0.440 ***
		Psychosom	92.57 [89.15; 95.18]	52.76 [48.85; 56.65]	

Note: The table entries in the “When having Other Outcome” columns represent the sensitivity and specificity to detect either outcome when the other outcome is present. The Kendall tau-b Correlation Coefficient represents the correlation between the symptoms of outcome pairs. Abbreviations: HRQoL, health-related quality of life; Psychosom, at least three psychosomatic complaints per week. ** *p* < 0.01, *** *p* < 0.001.

**Table 4 behavsci-13-00643-t004:** Predictors of elevated symptoms of anxiety, elevated depressive symptoms, and low HRQoL self-reported by adolescents in the second and the third year of the pandemic, with the number of self-reported weekly psychosomatic complaints included as an independent predictor.

Outcome	Gender	COP-S	Intercept ^#^	Age ^#^	Sociodemographic Condition ^#,†^	Parent’s Burden ^#,‡^	Child’s Burden ^#,§^	Family Climate ^#^	Psychosomatic Complaints ^#^	Model Fit (Nagelkerke’s R^2^)
Symptoms of Anxiety	Boys	2021	0.052 ***						1.582 *** [1.444; 1.731]	0.227
		2022	0.047 ***			1.870 ** [1.285; 2.741]		1.697 * [1.085; 2.647]	1.465 *** [1.341; 1.595]	0.218
	Girls	2021	0.024 ***				1.130 ** [1.041; 1.243]		1.565 *** [1.452; 1.692]	0.295
		2022	0.011 **				1.177 ** [1.063; 1.289]	1.823 ** [1.275; 2.806]	1.633 *** [1.511; 1.768]	0.378
Depressive symptoms	Boys	2021	0.005 ***	1.165 ** [1.014; 1.347]				2.810 *** [1.657; 4.666]	1.522 *** [1.347; 1.706]	0.257
		2022	0.005 ***	1.178 ** [1.062; 1.315]					1.597 *** [1.421; 1.786]	0.205
	Girls	2021	0.001 ***	1.097 * [1.003; 1.207]			1.232 *** [1.097; 1.381]	2.475 *** [1.580; 3.760]	1.731 *** [1.552; 1.933]	0.415
		2022	0.004 ***					2.287 *** [1.495; 3.649]	2.099 *** [1.851; 2.380]	0.458
Low HRQoL	Boys	2021	0.000 ***	1.198 ** [1.098; 1.302]			1.360 *** [1.217; 1.516]	3.113 *** [2.038; 4.728]	1.569 *** [1.425; 1.722]	0.420
		2022	0.001 ***	1.150 *** [1.063; 1.254]		1.680 ** [1.120; 2.439]	1.303 [1.162; 1.463]	2.788 *** [1.774; 4.427]	1.600 *** [1.466; 1.761]	0.390
	Girls	2021	0.000 ***	1.120 *** [1.031; 1.220]	3.369 ** [1.349; 8.409]		1.547 *** [1.391; 1.757]	4.209 *** [2.636; 6.151]	1.741 *** [1.578; 1.912]	0.549
		2022	0.000 *	1.170 *** [1.079; 1.278]			1.477 *** [1.477; 1.322]	2.592 *** [1.666; 1.947]	1.782 *** [1.622; 1.947]	0.524

^#^ The table indicates logistic regression odds ratios with 95% confidence intervals for all significant independent variables, controlling for other predictors. * *p* < 0.05, ** *p* < 0.01, *** *p* < 0.005. ^†^ Single parenthood OR low parental education OR parental mental health problems (dichotomous). ^‡^ Extended parental burden due to the pandemic (dichotomous). ^§^ Child’s pandemic-related burden due to school, reduced contact with friends, and extended use of digital media.

## Data Availability

The data presented in this study are available from the corresponding author upon reasonable request.

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
