# Peer review of "Adolescent Mental Health during the COVID-19 Pandemic: The Interplay of Age, Gender, and Mental Health Outcomes in Two Consecutive Cross-Sectional Surveys in Northern Italy"

_behavsci, 2023, doi:10.3390/bs13080643_

Round 1

Reviewer 1 Report

Greetings to the authors.

I really enjoyed reading this manuscript! I think it makes a valuable contribution.

A few comments that I hope will be helpful.

Lines 77 to 87 seem to not flow as well as the rest of the introduction. I wonder, perhaps, if you would consider reformatting? I think a singular paragraph on longitudinal studies would be better.

Shouldn't REB information be provided in the study design and sample? I see it at the end, but I think it should be mentioned in the study design and sample section.

My main concern is the title relates to Gender (i.e., men, women, non-binary etc), but your analyses use Sex (i.e., male, female intersex). These terms are not interchangeable and need to be adjusted. I really like this resource: https://www.justice.gc.ca/eng/abt-apd/pgbap-pacsp.html 

Page 6 of 22 is quite choppy, not complete paragraphs, but rather stand-alone sentences. I think this could be adjusted.

Best wishes,

Reviewer 2 Report

I would like to congratulate the authors for the manuscript they present, but in my opinion it can be improved. My comments are organized below for your consideration. I hope my comments are useful for the study author(s) and editorial staff.

#1 Introduction:

- Line 88 – 113 – This is a self-citation, while I beleive that your expertise in this area is outstanding, in order to enrich the literature of your manuscript and to suport the diversity, I kindle advise you to cite articles by other distinctive scholars in this field.

#2 Materials and Methods:

- The authors should mention if they used any guidelines to write the present manuscript according to the type of study.

- The authors should clarify the sampling method.

# References

- Line 766 – 767 – The reference is not correct
